# Critical Batch Size Revisited: A Simple Empirical Approach to Large-Batch Language Model Training

**William Merrill   Shane Arora   Dirk Groeneveld   Hannaneh Hajishirzi**
Allen Institute for AI
`willm@allenai.org`

## Abstract

The right batch size is important when training language models at scale: a large batch size is necessary for fast training, but a batch size that is *too large* will harm token efficiency. To navigate this tradeoff, McCandlish et al. (2018) suggest that a *critical batch size* (CBS), below which training will not substantially degrade loss, can be estimated based on the gradient noise scale during training. While their method has been adopted in practice, e.g., when training GPT-3, strong assumptions are required to justify gradient noise as a proxy for the CBS, which makes it unclear whether their approach should be trusted in practice, limiting its applicability. In this paper, we introduce a simple, empirical approach to *directly* measure the CBS and show how the CBS evolves over training. Applying our approach to the OLMo models, we find that CBS is near 0 at initialization, increases rapidly at first, and then plateaus as training progresses. Furthermore, we find that this trend holds across different model sizes (1B and 7B), suggesting CBS from small training runs can inform larger-scale training runs. Our findings about how the CBS changes over training motivate *batch size warmup* as a natural way to reliably train language models at large batch size: start the batch size small and increase it as the CBS grows. To validate this claim, we use batch size warmup to train OLMo 1B to slightly better loss than the original training run with 43% fewer gradient steps. This shows how our framework can be applied to reliably train language models at larger batch sizes, increasing data parallelism without compromising performance.

## 1  Introduction

Increasing the throughput of training is important for training large models. A natural way to increase throughput is by increasing data parallelism, i.e., increasing the *batch size* used during training so that more data can be processed at once and the number of sequential gradient steps can be decreased. However, naively picking a very large batch size can degrade the performance achieved by a fixed token budget, as larger batches can show diminishing returns in their ability to estimate the population gradient. Thus, in order to confidently train language models with higher token throughput, it is important to develop theoretical and empirical understanding of large-batch training.

One fundamental concept for large-batch training methodology is the following:

> **Critical Batch Size Hypothesis (McCandlish et al., 2018)**
>
> There is some critical batch size $B^*$ up to which increasing the batch size (and appropriately modifying the learning rate) approximately preserves the loss trajectory as a function of tokens trained, but, above which, the loss trajectory degrades.

If such a CBS $B^*$ exists (and we can measure it), it represents a reasonable balance between efficiency and performance (i.e., loss) and is thus a practically useful batch size at which to train. Working in a

simplified theoretical setup, McCandlish et al. (2018) derive a correspondence between the CBS and the *gradient noise scale*, i.e., the variance of the per-example gradients from the training distribution. They suggest that an estimator for the gradient noise scale should be used in practice as a proxy for the CBS, and this can in turn be used to set the batch size for large-scale pretraining runs. The noise scale also appears to have been adopted in practice as a proxy for the CBS, having been mentioned explicitly in the GPT-3 technical report (Brown et al., 2020), and inspiring a flurry of methodological innovations for better noise scale estimation (Gray et al., 2023, 2024).

While appealing, the link between the gradient noise and the CBS requires several strong assumptions to justify: specifically, it assumes the SGD optimizer and that gradients are well-conditioned (cf. Section 2). Thus, it is unclear whether the noise scale should be a meaningful proxy for the CBS for language model pretraining in practice, where the Adam optimizer is often used and the optimization may not be well-conditioned. To this end, we aim to address the following practical questions that remain for effectively leveraging the CBS viewpoint to train language models at larger batch sizes:

1. How can we measure the CBS cheaply with minimal assumptions before launching a pretraining run?

2. How does the CBS change over the course of pretraining and as a function of model size?

3. Having measured the CBS, how should we adapt the batch size, learning rate, and other parameters over the course of a pretraining run?

This work documents our attempts to answer these questions in order to operationalize the CBS for large-batch training. We focus our investigation on the OLMo models (Groeneveld et al., 2024; OLMo et al., 2025), due to their open weights and data, making the following contributions:

1. First, we introduc an empirical method to directly measure the CBS via **branched training**. Our method avoids strong assumptions needed to justify the noise scale method from prior work. This lets us trust it more than the noise scale method, which we find unreliable.

2. We use our method to study how our CBS measurement changes over the course of training, finding it improves rapidly initially but than flattens off. Further, we find that the CBS not depend strongly on model size, in line with past findings using different methodology (Zhang et al., 2024).

3. Our knowledge of the local CBS across training checkpoints suggest a natural **batch size warmup** strategy for large batch training: begin training with a small batch size and double it whenever the CBS increases sufficiency. We use this strategy to train 1B parameter models with 43% fewer gradient steps without degrading (and, in fact, slightly improving) final loss.

Overall, our empirical framework for measuring and leveraging the CBS provides simple and principled methodology for improving the efficiency of large-scale training runs and addressing other fundamental questions in the science of language model pretraining.

## 2 Background: Estimating CBS via Gradient Noise Scale

Past empirical work has aimed to measure the CBS by launching many training runs to the same target loss, which is expensive (Zhang et al., 2019, 2024), or by using the gradient noise scale as a proxy (McCandlish et al., 2018; Gray et al., 2023, 2024). In contrast, we will introduce a new method that uses a small amount of additional training to estimate the CBS, which is less expensive than launching many full training runs and does not make any strong assumptions like the noise scale. Before introducing our method, we review the noise scale framework used to estimate the CBS in prior work and the underlying assumptions it relies on.

McCandlish et al. (2018) suggest that the CBS can be measured in terms of the gradient noise scale, i.e., the variance of the gradients within a batch. Concretely, their recommendations to measure the CBS and adapt the learning rate are as follows:

> **Existing Method: Noise Scale Proxy for CBS (McCandlish et al., 2018)**
>
> Let $G$ be the full gradient and let $\Sigma$ be the covariance matrix for the gradient across data examples. We first compute $\mathcal{B}_{\mathsf{simple}}$ as a proxy for the CBS (using an efficient statistical estimator):
>
> $$\mathcal{B}_{\mathsf{simple}} = \frac{\mathrm{tr}(\Sigma)}{\|G\|^2}.$$
>
> We set the modified batch size to $\mathcal{B}_{\mathsf{simple}}$ and *linearly* scale the learning rate $\eta^*$:
>
> $$B^* = \mathcal{B}_{\mathsf{simple}} \tag{1}$$
>
> $$\eta^* = \frac{B^*}{B} \cdot \eta. \tag{2}$$

This viewpoint is attractive due to its simplicity and tractability, and it has inspired improved methods for estimating the noise scale (Gray et al., 2023, 2024). However, the link between noise scale and CBS requires several strong assumptions to justify, which we will argue motivates revisiting other approaches for measuring the CBS. McCandlish et al. (2018) consider training a model on a loss surface where the loss landscape is well approximated by its second-order Taylor expansion. The first crucial assumption in their analysis is that optimizer used to decrease the loss is SGD:

> **Assumption 1: SGD Optimizer (McCandlish et al., 2018)**
>
> The step taken to reduce the loss is a noisy estimate of the true gradient, computed via $B$ samples.

This assumption may seem benign, but it is worth noting that it is *not* typically met in practice, as LMs are trained with the Adam optimizer (Kingma and Ba, 2017). Moreover, by analyzing training dynamics in terms of stochastic differential equations (Li et al., 2021), Malladi et al. (2022) argue theoretically that Equation (2) is an appropriate scaling rule for SGD, but not for Adam, where a *square-root* scaling rule is more principled. The square-root scaling for Adam is also supported by the theoretical analysis of Li et al. (2024, Equation 4). Thus, when training with Adam, it seems that the linear scaling rule assumed by McCandlish et al. (2018) should not apply.

A more fundamental issue is that McCandlish et al. (2018) also require another strong assumption to derive the noise scale method for estimating the CBS. In general, their noise-scale-based estimate of the CBS has a more complex form than $\mathcal{B}_{\mathsf{simple}}$ involving the Hessian $H$:

$$\mathcal{B}_{\mathrm{noise}} = \frac{\mathrm{tr}(\Sigma H)}{G^\top H G}.$$

In order to justify that $B^*$ can be computed as $\mathcal{B}_{\mathsf{simple}}$, McCandlish et al. (2018) assume:

> **Assumption 2: Well-Conditioned Optimization (McCandlish et al., 2018)**
>
> The Hessian $H$ is a multiple of the identity matrix. It follows that $\mathcal{B}_{\mathrm{noise}} = \mathcal{B}_{\mathsf{simple}}$.

This is a strong assumption required to justify using $\mathcal{B}_{\mathsf{simple}}$ as a proxy for the CBS because computing Hessians would be too expensive to be practice. McCandlish et al. (2018) suggest informally that, without this assumption, $\mathcal{B}_{\mathsf{simple}}$ *may* still be correlated with $\mathcal{B}_{\mathrm{noise}}$, but it is not obvious why this should be the case. Even if this is true, it still poses a real problem for the noise scale methodology, since the goal of the method is to produce an *absolute* measure of $B^*$. It is unclear for practitioners what coefficient should be used to translate $\mathcal{B}_{\mathsf{simple}}$ to $B^*$—and, more fundamentally, whether it is even valid to assume that such a coefficient exists.

## 3  Our Method: Measuring the CBS via Local Branched Training

As discussed in Section 2, using the gradient noise scale $\mathcal{B}_{\mathsf{simple}}$ as a proxy to estimate the CBS relies on several strong assumptions. In light of this, it unclear whether we can trust the gradient noise scale as a proxy for CBS. We thus argue that we should instead aim to measure the CBS *directly*, without the need for any strong assumptions to justify an indirect proxy. After introducing our measurement

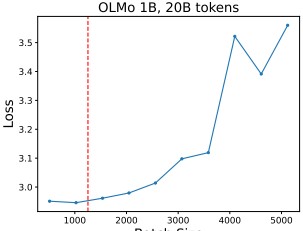 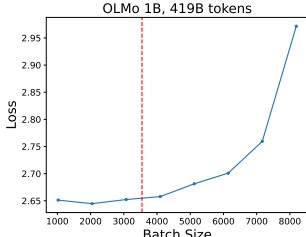 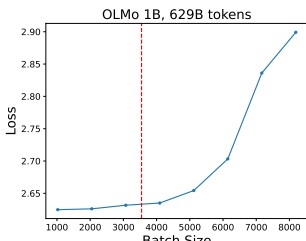

Figure 1: Smoothed final loss after branched training at particular checkpoints, with $B^*$ shown as the dotted red line. Each point represents the loss achieved by a specific branched training run after 2B tokens. Our method detects the point at which loss starts to increase, heuristically tolerating noise within $\epsilon = 0.01$. These plots show how this plays out for three particular checkpoints; see Appendix A for loss curves for all checkpoints.

approach in this section, we will show in Section 4 how these measurements can be applied to train language models to the same (or better) target loss with fewer gradients steps.

## 3.1 Method

We introduce a simple **branched training** method that directly approximates the CBS by launched branched training runs from a checkpoint, which allows us to identify $B^*$ as the largest batch size that does not degrade in loss relative to smaller batch sizes as visualized in Figure 2. To make this tractable, we train only for a fixed token budget $\Delta$, assuming that if $B^*$ recovers in loss by $\Delta$, its loss will continue to match smaller batch sizes onwards as well. This allows us to estimate the CBS with only a small amount of additional training (controlled by $\Delta$). Further, as we will find later in Figure 2, the CBS trend remains consistent across model sizes, so CBS measurements with small models could be used to inform large-scale training runs.

> ### Our Method: Branched Training to Measure CBS
>
> Given a training checkpoint with original batch size $B$ and learning rate schedule $\eta$, we aim to measure the critical batch size $B^*$. Let $f(\eta)$ be the learning rate scaling rule: $f(k) = k$ for SGD and $\sqrt{k}$ for Adam. We create several training branches with modified batch size $k \cdot B$ and learning rate $f(k) \cdot \eta$ and train for a small number of tokens $\Delta$ to get loss $L_k$—following standard practice, we take $L_k$ to be the smoothed loss. We then define $k^*$ as the maximum $k$ such that, for all $k < k^*$, $L_{k^*} \leq L_k + \epsilon$, where $\epsilon$ is a tolerance parameter for "similar" losses. We then define the CBS $B^*$ and scaled learning rate $\eta^*$ as
>
> $$B^* = k^* \cdot B$$
> $$\eta^* = f(k^*) \cdot \eta.$$

Our method will empirically estimate the largest batch size $B^*$ at which the local optimization trajectory recovers roughly to its original loss after $\Delta$ steps. In contrast to the strong assumptions needed to justify the gradient noise scale, the only (weak) assumption our method relies on is:

> ### Assumption 3: Local Recovery
>
> If the loss achieved by batch size $B^*$ recovers to match the loss with batch size $B < B^*$ after training for $\Delta$ tokens, the loss trajectories will remain the same beyond $\Delta$ as well.

**Implementation Details.** Our method requires specifying two parameters: the window size $\Delta$ and the loss tolerance $\epsilon$. We also apply smoothing to the loss to reduce noise. In more detail:

1. **Window Size.** Because the optimizer state must update when the batch size is changed, we expect an immediate spike in the loss after adjusting the batch size. The window size $\Delta$ represents the number of steps we are willing to wait for the loss to recover from this bump. The CBS measurements could, in principle, depend on $\Delta$, with larger values of $\Delta$

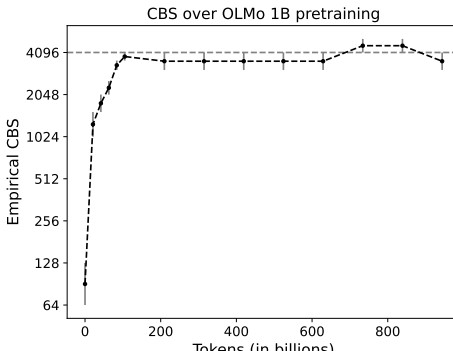
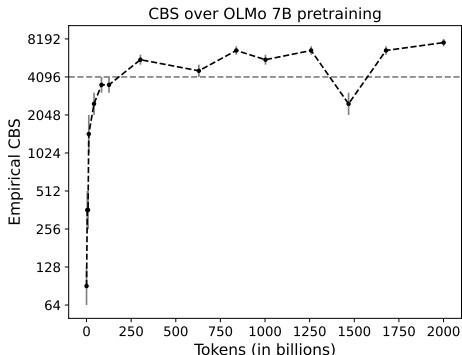

Figure 2: CBS over training for OLMo 1B and 7B, measured in documents (4096 tokens per document). The qualitative trend is similar across both model sizes. The CBS starts near 0, grows rapidly but diminishingly, and plateaus around 4096.

       potentially producing larger CBS estimates. We set $\Delta$ to 2B tokens, which we take as a small, conversative window size relative to our overall pretraining budget of 600B tokens.
2. **Loss Tolerance.** Viewing loss as a function of batch size multiplier $k$ (cf.Figure 1), we need a way to determine whether the loss at $k^*$ has increased relative to all $k < k^*$. We operationalize this with a tolerance parameter $\epsilon$, which we set to 0.01 arbitrarily. In principle, tolerance could be set in a more principled way using a statistical test in future work.
3. **Loss Smoothing.** Pretraining loss is noisy at the batch level, so, following standard practice, we apply exponentially moving average smoothing with parameter $\alpha$ set to 0.5.

It is worth comparing our method to other work that empirically measures the CBS. Our method of training for a fixed token budget $\Delta$ and measuring the change in loss can be understood as the dual view to measuring the number of steps required to achieve a target loss, which has been used in prior work (Zhang et al., 2019, 2024). Reformulating the measurement in this way has the nice property that the training budget can be fixed in advance. Under the local recovery assumption, it also allows us to train for only $\Delta$ tokens, which means we do not have to launch full training runs for each batch size. Finally, unlike Zhang et al. (2024), we apply our method at various checkpoints throughout training, whereas they apply it only from initialization. This means we can estimate the *local CBS* at a specific point in training rather than just the *global CBS*.

## 3.2 Experimental Setup

We aim to measure the CBS over the course of model training and the role of model size. Because our method requires pretraining checkpoints and access to the pretraining data, we use the OLMo 1B and OLMo 7B models for our experiments (OLMo et al., 2025), whose pretraining data is openly available (Soldaini et al., 2024). For each model, we take a variety of checkpoints over the course of training, allowing us to assess how the CBS changes over the course of training; see Appendix A for more details. We also compute the noise scale across training checkpoints using the estimator proposed by McCandlish et al. (2018) to assess whether it is a valid proxy for the CBS we measure.

We define the interval for the CBS at each checkpoint by choosing $B^*$ (as defined in Section 3) as a lower bound, and by picking the least $k > k^*$ as an upper bound. The plotted point represents the geometric mean of these interval endpoints. We measure the CBS in documents, with a pretraining sequence length of 4096 tokens per document.

## 3.3 Results: CBS Over Training and Across Model Sizes

Figure 2 shows the CBS $B^*$ measured via Section 3.1 across training checkpoints for OLMo 1B and 7B. The CBS increases over training in a similar way for both model sizes: the CBS starts near 0, grows rapidly within the first 50k tokens, and then plateaus around 4096.

**Impact of Model and Data Size.** Prior work has suggested that the CBS is largely independent of model size, scaling primarily with data size (Zhang et al., 2024; Bergsma et al., 2025). This is largely

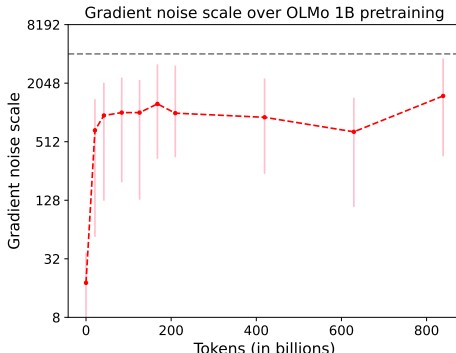 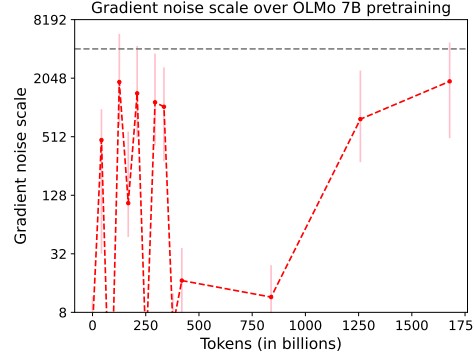

Figure 3: Gradient noise scale for OLMo 1B and 7B computed via the estimator of McCandlish et al. (2018) with 95% confidence intervals; details in Appendix B. The gradient noise scale underestimates the CBS (cf. Figure 2) and the qualitative trend does not clearly match, especially for OLMo 7B.

consistent with our findings, as the CBS curves like qualitatively similar at the 1B and 7B scales. In addition, the fact that the CBS grows over training suggests that the "aggregate" CBS should also increase as we train on more data since the average CBS over the course of training should increase. We elaborate on this in Appendix D: while CBS growth over training predicts that aggregate CBS should increase with data size, it is unclear whether the CBS growth pattern we observe would predict the aggregate CBS $\propto \sqrt{D}$ scaling law found in prior work (Zhang et al., 2024; Bergsma et al., 2025).

**Comparison with Gradient Noise Scale.** As discussed in Section 2, the gradient noise scale has been proposed as a proxy to measure the CBS (McCandlish et al., 2018), though this connection relies on strong assumptions to justify. We thus empirically compare our measurement of the CBS to the gradient noise scale. Figure 3 shows that, for both OLMo 1B and 7B, the gradient noise scale underestimates the CBS by several orders of magnitude. Furthermore, especially for OLMo 7B, the qualitative trend does not match the CBS. For OLMo 1B, the qualitative pattern is more similar. However, since this similarity is not found for both models, we conclude that, in general, the noise scale cannot be used reliably as a proxy for the CBS.

**Motivating Batch Size Warmup.** A central takeaway from these results is that the CBS starts near 0, grows rapidly, and then plateaus. This suggests that *batch size warmup*, where the batch size is dynamically increased over the beginning of a training run, is a natural way to increase the effective batch size for most of training while avoiding training with a batch size above the CBS. In the next section, we will discuss our implementation and validation of this idea.

## 4 Application: Batch Size Warmup for Larger Batch Training

A straightforward way to speed up training is to increase the batch size, and, as long as the new batch size is less than the CBS, we can be confident that the loss will be minimized about as effectively as before. Thus, we can leverage knowledge of the CBS to speed up training.

Furthermore, our local knowledge of the CBS can be leveraged to do this more effectively than if we only had a global sense of the CBS. If we simply increased a fixed batch size, this would mean that we are training with a batch size that is too large for a short period at the beginning of training, which could potentially destabilize training or degrade final performance. To get around this, we can use our measurement of the local CBS to "warm up" the batch size. We will train at a smaller batch size for the beginning of training when the CBS is small, and then switch to a larger batch size once the CBS grows large enough. More generally, we can aim to double the batch size whenever we determine the CBS has doubled. This should allow us to benefit from training at a larger batch size for *most* of training without training at a batch size that is too large at the beginning of training, which could degrade the final loss achieved by the training run.

### 4.1 Methodology for Batch Size Warmup

Given an existing training run (at a small batch size), we aim to adapt it to achieve the the same final loss with fewer overall gradient steps (i.e., a larger batch size for most of training). Assuming an original fixed batch size $B$ and base learning rate $\eta$, our method for training with *batch size warmup* is as follows. First, we use branched training CBS method (Section 3) to measure $B_t^*$, the CBS after training for $t$ tokens, using an existing checkpoint. When training a new model, we set the batch size and and learning rate as follows:

---

**Batch Size Warmup**

1. At $t = 0$, initialize the batch size to $B_0 = B$ and base learning rate to $\eta_0 = \eta$.
2. After training for $t$ tokens, if we determine that the CBS exceeds the current batch size ($B_t^* > 2B_t$), we double the current batch size and update the base learning rate following the square-root scaling rule (cf. Section 3):

$$B_{t+1} = 2B_t$$
$$\eta_{t+1} = \sqrt{2}\eta_t.$$

---

This method will ensure that the batch size will increase over training *safely*, i.e., never exceeding the CBS. Since we found that the CBS increases quite rapidly at the beginning of training, this method will double the batch size twice early in training, reaching a maximum batch size of 4096 by 503B tokens. This means that we will effectively train at a larger batch size for most of training compared to the original small-batch run. But, crucially, we can be confident that the batch size will never increase our batch size above the CBS. Thus, we expect the final loss will be comparable to that of the original training across thet training trajectory.

**Implementation Details.** The main implementation question is how to operationalize checking $B_t^* > 2B_t$ in order to double the batch size. In practice, we do this heuristically based on the measurements in Figure 2: since we only double the batch size twice over training, this involves just picking two thresholds. In addition, this process could be automated for future training runs using online measurements of the CBS paired with some kind of curve fitting or statistical test.

**Design Choices.** As defined above, our method uses a square-root scaling rule, which is well-motivated for Adam (Malladi et al., 2022), but in principle a linear scaling rule could also be used. As batch size warmup only modifies the base learning rate, it is compatible to overlay with existing learning rate schedules (in practice, the models we use follow a cosine schedule). Finally, we choose to only increase the batch size at powers of two because it is not clear that having a more precise match to the CBS beyond the order-of-magnitude level would be particularly useful and because the OLMo codebase requires the number of GPUs $g$ divides the batch size. Thus, doubling is convenient because it easily guarantees the new batch size remains a multiple of $g$. However, in principle one could update more frequently, e.g., by setting updating the batch size the largest multiple of $g \leq B_t^*$.

**Connection to Existing Methods.** Our batch size warmup method is similar to preliminary experiments by McCandlish et al. (2018, Appendix D) using a dynamic batch size on the SVHN dataset. However, they use their noise scale method (Section 2) to set the batch size, which is potentially unreliable, and only consider small-scale classification. In the context of image classification, prior work explored replacing learning rate decay with increasing the batch size (Smith et al., 2018). This is conceptually related to our batch size warmup approach in that it leverages scaling rules between batch size and learning rate to achieve larger batch training. However, our method can apply on top of existing learning rate schedules, can generalize to Adam (vs. SGD), and, most crucially, ensures that the batch size number exceeds the critical batch size.

### 4.2 Experimental Setup

We evaluate the viability of training with batch size warmup via the following training runs. For all models, we use the default OLMo 1B pretraining hyperparameters unless stated otherwise.

1. **Batch Size Warmup:** We train OLMo 1B with our batch size warmup method as detailed in Section 4.1. We initialize the batch size to 1024. Based on a manual reading of the CBS

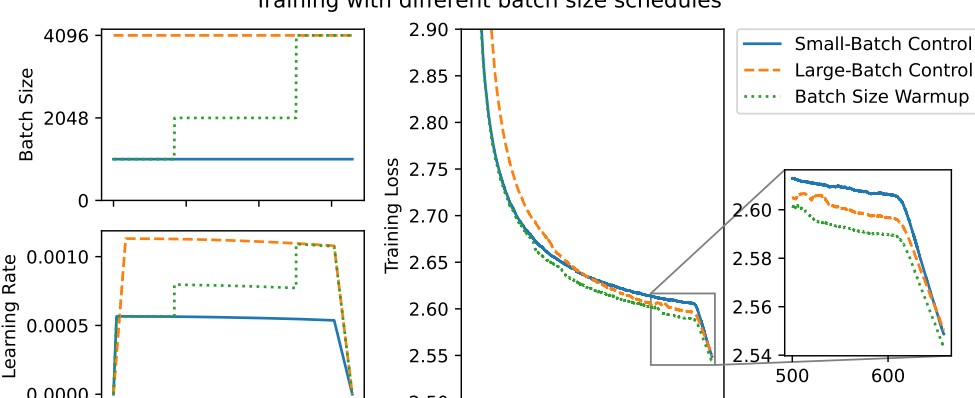

Figure 4: Batch size schedule (left, top), learning rate schedule (left, bottom) and training loss (right) for the pretraining of an OLMo model with different batch size schedules. Training loss is smoothed by taking the moving average over the past 10B tokens.

| Method | ↓ PT Loss | ↓ MT Loss | ↑ Grad. Steps Saved |
|---|---|---|---|
| Batch Size Warmup (Ours) | **2.5891** | **2.5433** | 43% |
| Small-Batch Control | 2.6057 | 2.5486 | 0% |
| Large-Batch Control | 2.5962 | 2.5506 | **75%** |

Table 1: Loss after pretraining (PT) and mid-training (MT) for each run, averaged over the past 10B tokens and the percentage of gradient steps saved vs. the small-batch control (including annealing steps). Both before and after annealing, batch size warmup slightly outperforms both controls in loss.

measurements in Section 3, we determine that the CBS reaches 2048 by 168B tokens and 4096 by 503B tokens.[1] Our method thus doubles the batch size at each of these points.

2. **Small-Batch Control:** We train OLMo 1B with batch size $B = 1024$ and base learning rate $\eta = \sqrt{2} \cdot 0.0004$.[2] We expect that our batch size warmup method should be able to achieve similar final loss to the small-batch control with fewer gradient steps.

3. **Large-Batch Control:** We train OLMo 1B with a fixed large batch size of $B = 4096$ and base learning rate $\eta = 2\sqrt{2} \cdot 0.0004$. Since its batch size will exceed the critical batch size for the initial part of training, we expect that the large-batch control will show degraded loss compared to our batch size warmup method.

**Evaluations.** Language model training consists of two steps, pre-training and mid-training (OLMo et al., 2025). We evaluate training loss as well as several measures of out-of-distribution loss at the end of the pre-training stage as well as after the mid-training stage.

- **Loss after Pretraining.** We evaluate the loss at the end of the pre-training stage for all models. The original training run for OLMo 1B ran for 4T tokens, so we do not have the

---

[1]These thresholds were determined from preliminary CBS measurements at an earlier stage of the project. Our measurements changed slightly after this point, but we deemed that it was not worth it to restart the expensive training runs because we do not think the results should be sensitive to a precise choice of threshold. Going forward, it could be useful to establish systematic methodology for choosing batch size warmup thresholds given the CBS measurements.

[2]The default hyperparameters for OLMo 1B use a batch size of 512 and learning rate of 0.0004. Since we chose an initial base size of 1024 for our experiments, we adapted the learning rate appropriately via the square-root scaling rule (Malladi et al., 2022).

| Method | ↓ Task BPB | | ↓ C4 | | ↓ Pile | |
|---|---|---|---|---|---|---|
| | PT | MT | PT | MT | PT | MT |
| Batch Size Warmup (Ours) | 1.0316 | 1.0076 | **2.8049** | **2.7597** | **2.1916** | 2.1521 |
| Small-Batch Control | **1.0112** | **0.9999** | 2.8196 | 2.7622 | 2.2073 | **2.1471** |
| Large-Batch Control | 1.0571 | 1.01927 | 2.8107 | 2.7658 | 2.1996 | 2.1586 |

Table 2: According to the method from Bhagia et al. (2024), we evaluate downstream performance via BPB on downstream tasks, as well as cross-entropy loss on two held-out sets, C4 and the Pile, both after pretraining and after mid-training. Batch-size warmup generally performs comparably or better compared to the small-batch control, suggesting it does not degrade downstream performance.

        resources to replicate this full training run. Instead, we pre-train for 608B tokens using the original learning rate schedule for the longer run.

- **Loss after Mid-Training.** We report loss at the end of mid-training stage, which more closely reflects how these checkpoints are used in language model training. Specifically, starting from final pretraining checkpoint, we linearly anneal the remaining learning rate down to 0 for 50B tokens, keeping the batch size fixed at its final value (Figure 1). Because mid-training is a part of the standard language modeling pipeline, we take the loss after mid-training to represent the loss that would be achieved by these training runs in a practical context. Furthermore, OLMo et al. (2025) suggest that this kind of learning rate annealing can induce significant gains in loss for partial pretraining runs. We thus also take the loss after mid-training loss as a proxy for how these runs would compare if we were to fully train them for the full learning rate schedule.

- **Out-of-Distribution Losses.** To measure performance beyond the training loss, we also track three kinds of out-of-distribution losses. The first two are straightforward cross-entropy losses on common pretraining datasets, C4 (Dodge et al., 2021) and The Pile (Gao et al., 2020). The third out-of-distribution loss follows the method from Bhagia et al. (2024), which argues for computing the loss (in bits-per-byte; BPB) on the correct answers of multiple popular question-answering datasets such as ARC-Easy, ARC-Challenge, MMLU, etc. For the full list of datasets used, see Appendix E.

### 4.3 Batch Size Warmup Results

We compare batch size warmup to the small and large-batch controls at two points: first, after the pretraining stage and, second, after the mid-training stage that anneals the remaining learning rate down to 0. After pretraining, batch size warmup reaches lower loss than both controls, as shown in Figure 2 and Table 1, outperforming the small-batch control in loss by a margin of 0.0166. After mid-training, batch size warmup still slightly outperforms the small-batch control in loss, this time by a smaller margin of 0.0053. The large-batch control now achieves the worst overall loss. Overall, batch size warmup slightly exceeds the small-batch control in loss while using 43% fewer gradient steps. In contrast, the large-batch control uses 75% fewer gradient steps, but its final loss degrades compared to the small-batch control. Thus, we find batch size warmup is a reliable method to train with fewer gradient steps without degrading final loss.

Beyond the final loss, we also evaluate whether batch size warmup leads to similar downstream performance as small-batch training using our downstream-task BPB evaluation as well as validation loss on C4 and the Pile. As shown in Table 2, we find that BPB on validation sets is broadly competitive with the small-batch control, performing best on three out of four conditions considered. With the caveat that precisely measuring the downstream impact of pretraining decisions is difficult, this suggests that, just as batch-size warmup does not degrade final loss, it should not degrade downstream measures of performance either.

## 5 Conclusion

In this work, we introduced a simple empirical method for estimating the CBS throughout language model pretraining runs, which can be used to increase batch size (and thus effective token throughput) for large scale training runs without sacrificing performance. As we discussed, the existing noise

scale method (McCandlish et al., 2018) for estimating the CBS requires strong assumptions to justify, which our approach avoids. We used our method to study the evolution of the CBS during training for the OLMo models, finding that CBS increases monotonically but diminishing over the course of training, and that CBS does not seem to depend on model size, in line with prior work (Zhang et al., 2024). Guided by these findings, we showed that our measurements could be used to pick a **batch size warmup** schedule that enables larger batch training without harming final training loss. We take these results to demonstrate the validity and utility of our CBS measurement approach, and believe our framework could be useful for increasing the efficiency of future large-scale pretraining efforts.

There are several details and extensions of our method that would be interesting to explore going forward. First, it would be interesting to carry out a more systematic analysis of the impact of the hyperparameter $\Delta$ on CBS measurements: how sensitive is the method to $\Delta$ and do different values for $\Delta$ potentially bias the measurement? It would also be interesting to further investigate the conditions under which noise scale might be a meaningful CBS proxy and used for batch size warmup. When applying our CBS measurements to picking a batch size warmup schedule, we manually picked doubling threshold based on the CBS. Going forward, it would be useful to establish a systematic way to set threshold for increasing the batch size given CBS measurements. There are also other potential methodological improvements, such as removing the arbitrary power of 2 constraint and estimating the CBS in an online fashion. Overall, these improvements would allow batch size warmup to be applied more robustly and easily across different pretraining setups.

## Acknowledgments

We thank Joel Hestness, Sadhika Malladi, and Ananya Harsh Jha for discussions.

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

# A    CBS Measurement Details

Our empirical method for measuring the CBS Section 3 is, in principle, sensitive to the choice of checkpoints and batch size multipliers. We therefore document the checkpoints and multipliers we used here.

**OLMo 1B.** When measuring the OLMo 1B CBS with branching, we set the base batch size to 1024 and the base learning rate to $0.0004 \cdot \sqrt{2}$, reflecting the default batch size of 512 and learning rate of 0.0004 in the OLMo codebase under a square-root scaling rule (Malladi et al., 2022). We then chose the following checkpoints and multipliers $k$:

1. Step 0: $k$ ranging over 0.0625, 0.125, 0.25.
2. Steps 10K, 20K, . . ., 50K: $k$ ranging over 0.5, 1, . . ., 5.
3. Steps 100K, 150K, . . . , 450K: $k$ ranging over 1, 2, . . ., 8.

Figure 5 shows loss vs. batch size plots for all checkpoints of OLMo 1B.

**OLMo 7B.** We set the base batch size to 1024 and the base learning rate to 0.0003, as specified in the OLMo codebase. We then chose the following checkpoints and multipliers $k$:

1. Step 0: $k$ ranging over 0.0625, 0.125, 0.25.
2. Steps 1K, 2K, 3K: $k$ ranging over 0.25, 0.5, 1, 2, 3, 4.
3. Steps 10K, 20K, 30K: $k$ ranging over 1, 2, 3, 4, 5.
4. Steps 72K, 150K, 200K, 239K, 300K, 350K, 400K, 477K: $k$ ranging over 1, 2, 3, 4, 5, 6, 7, 8.

Appendix A shows loss vs. batch size plots for all checkpoints of OLMo 7B. These checkpoints were chosen manually as we developed this project. Over the course of our experimentation, we launched many additional runs beyond the ones discussed above. Since the choice of $k$ can influence the conclusions of our method, we filtered down the included runs to make the choice of $k$ systematic.

1B runs were launched on a single node of H100 GPUs, and 7B runs were launched on 8 nodes.

# B    Noise Scale Measurement Details

We use the gradient noise scale estimator proposed by McCandlish et al. (2018, Appendix A) to estimate the gradient noise scale. The method estimates the gradient noise scale using gradient norms at two different batch sizes $B_{\text{big}}$ and $B_{\text{small}}$ according to:

$$\mathcal{B}_{\text{simple}} \approx \frac{\mathcal{S}}{\|\mathcal{G}\|^2}, \text{where}$$

$$\mathcal{S} = \frac{\|G_{\text{small}}\|^2 - \|G_{\text{big}}\|^2}{1/B_{\text{small}} - 1/B_{\text{big}}}$$

$$\|\mathcal{G}\|^2 = \frac{B_{\text{big}}\|G_{\text{big}}\|^2 - B_{\text{small}}\|G_{\text{small}}\|^2}{B_{\text{big}} - B_{\text{small}}}.$$

We use large batch size $B_{\text{big}} = 64$ and small batch size $B_{\text{small}} = 1$.

It holds that $\mathbb{E}\left[\mathcal{S}\right] = \text{tr}(\Sigma)$ and $\mathbb{E}\left[\|\mathcal{G}\|^2\right] = \|G\|^2$. We thus average $\mathcal{S}$ and $\|\mathcal{G}\|^2$ over 4096 batches reduce variance and then return their ratio as our estimate of the noise scale $\mathcal{B}_{\text{simple}}$, using offline (i.e., unseen) data in each batch.

We estimate a confidence interval for $\mathcal{B}_{\text{simple}}$ in two steps. First, we estimate 95% confidence intervals for $\mathcal{S}$ and $\|\mathcal{G}\|^2$, assuming the data are exponentially[3] and normally distributed, respectively, based

---

[3]For the exponential distribution, we use approximate confidence interval under "Confidence Intervals" here: https://en.wikipedia.org/wiki/Exponential_distribution.

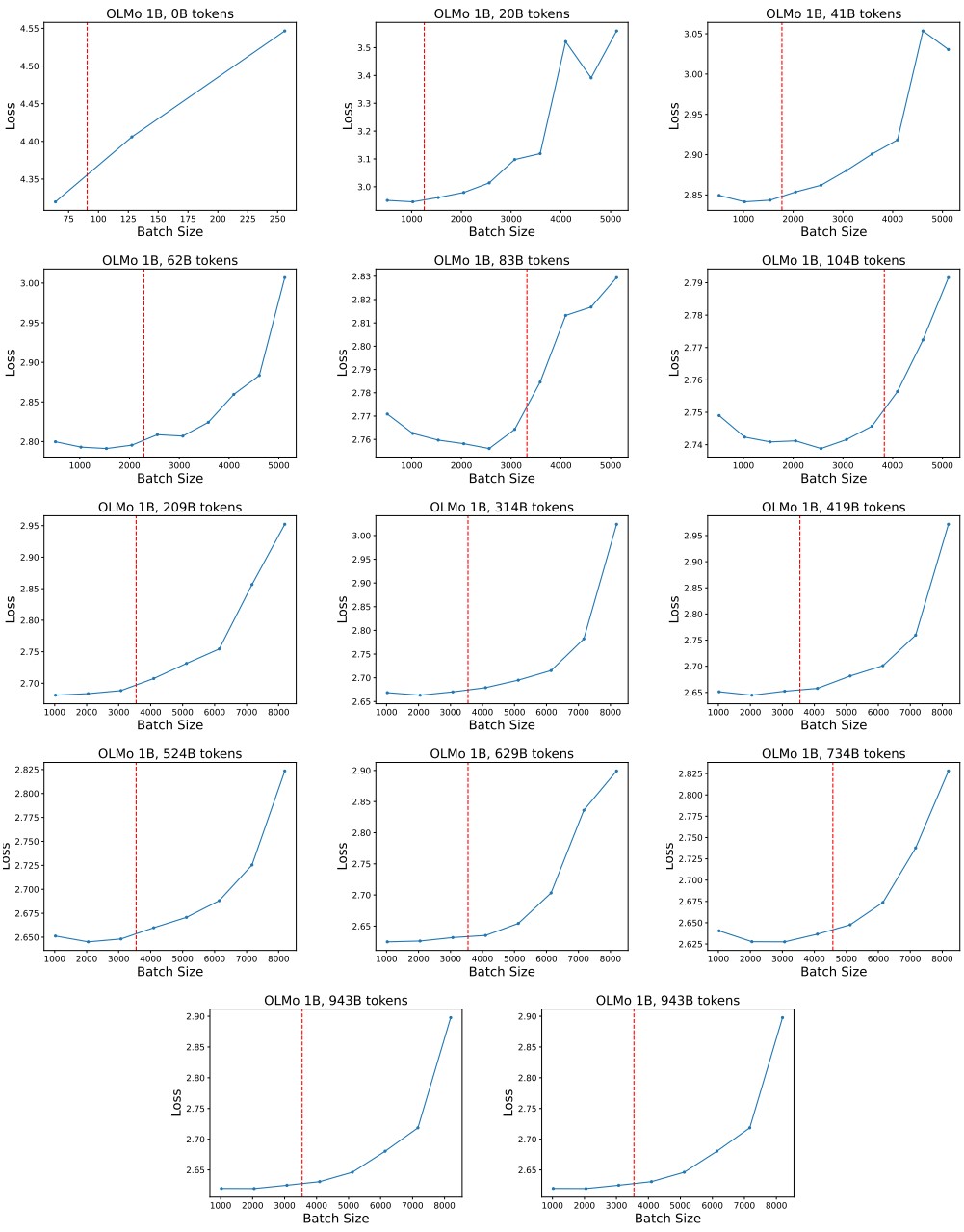

Figure 5: All loss vs. batch size plots for OLMo 1B. Overall, the red line moves to the right over time, showing that the CBS increases.

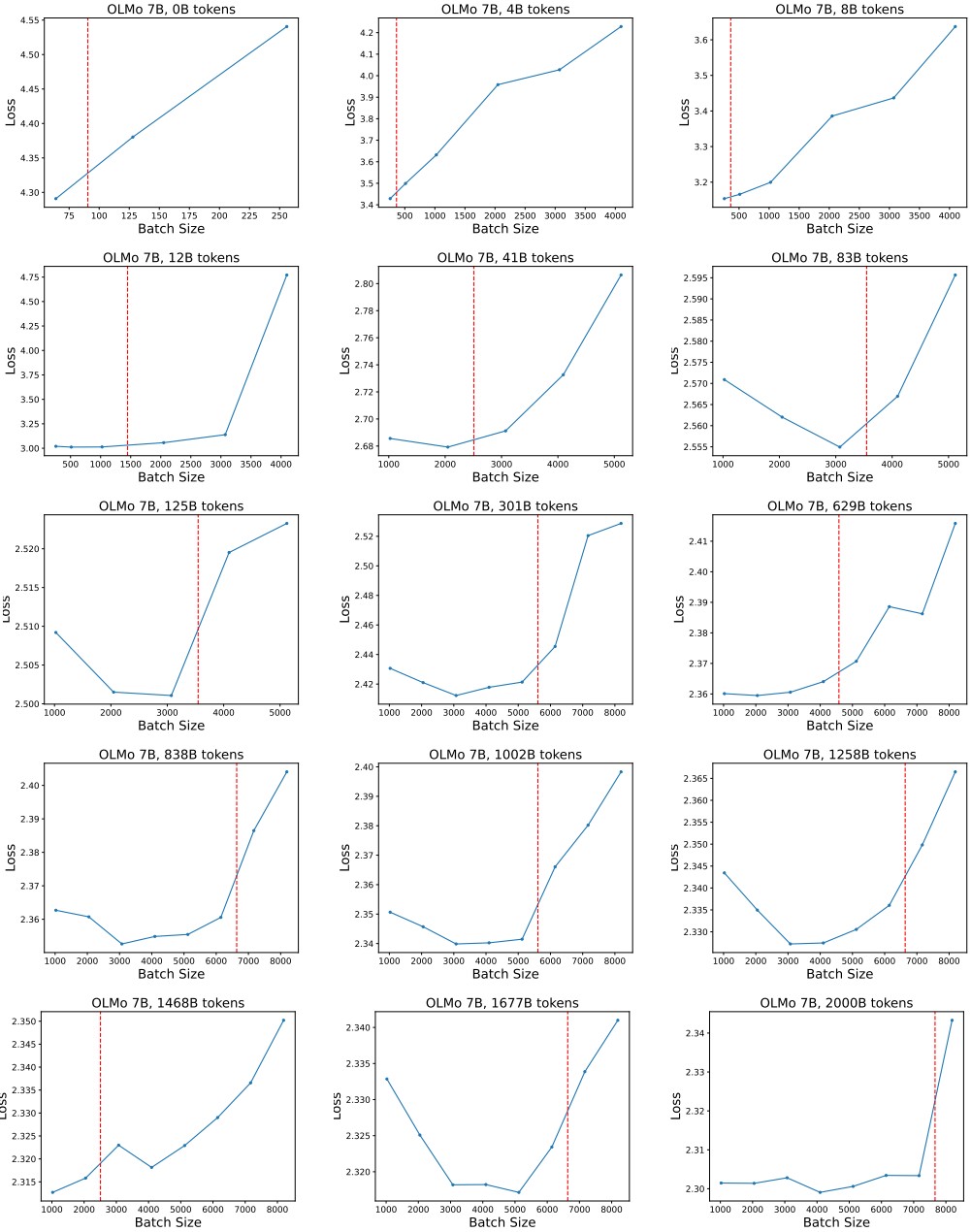

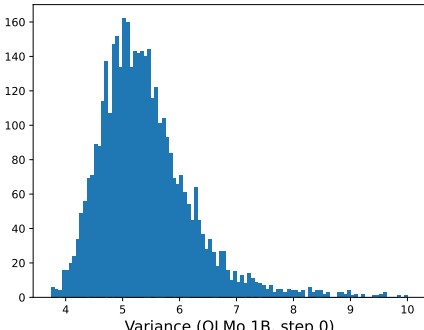
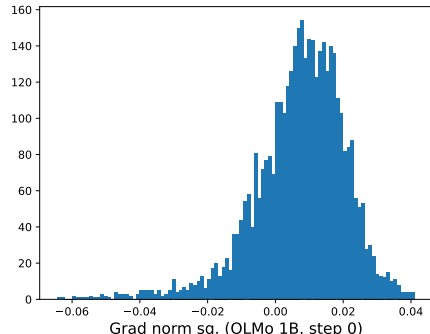

Figure 6: Representative histograms for $\mathcal{S}$ and $\|\mathcal{G}\|^2$, showing data from the 1st to 99th percentiles. The distribution for $\mathcal{S}$ is positive, leading us to use an exponential distribution, while the fact that some samples of $\|\mathcal{G}\|^2$ are negative motivates a normal distribution.

on manual inspection of their distributions (cf. Figure 6). We denote these intervals $[a_{\mathcal{S}}, b_{\mathcal{S}}]$ and $[a_{\|\mathcal{G}\|^2}, b_{\|\mathcal{G}\|^2}]$, respectively. We then define the confidence interval for $\mathcal{B}_{\text{simple}}$ as follows:

$$\left[ \frac{a_{\mathcal{S}}}{b_{\|\mathcal{G}\|^2}}, \frac{b_{\mathcal{S}}}{a_{\|\mathcal{G}\|^2}} \right].$$

If our estimates for $\mathcal{S}$ or $\|\mathcal{G}\|^2$ (or their lower or upper bounds) come out negative, we consider them to be 0.

The checkpoints considered for OLMo 1B are steps 0, 10K, 20K, 40K, . . ., 100K, 200K, . . . 400K. For OLMo 7B, we use checkpoints at steps 0, 10K, . . ., 40K, 60K, 70K, . . ., 100K, 200K, . . . 400K. The noise scale experiment for each checkpoint (for both the 1B and 7B models) was launched on a single GPU.

## C   License Information

The OLMo models (Groeneveld et al., 2024; OLMo et al., 2025) and pretraining code, which we use, are released under Apache-2.0 license. C4 (Dodge et al., 2021) is released under ODC-BY license. The Pile (Gao et al., 2020) is released under MIT license.

## D   Deriving CBS Scaling Laws: An Attempt

In this section, we explore whether our empirical fits for the critical batch size over training can be used to derive scaling laws for aggregate critical batch size that have been derived in prior work. These scaling laws assume we want to use a fixed batch size $B$ over training, and then train many different models to the same target loss. They then measure the critical $B^*$ up to which increases in batch size do not diminish token efficiency. The standard finding from such work is that CBS grows $\propto \sqrt{T}$, where $T$ is the total training budget in tokens. This is consistent with our finding that CBS increases over the course of training—moreover, we now seek to analyze whether this scaling law can be derived from our empirical measurements of local CBS. If so, this would provide converging evidence and a simpler method for fitting CBS scaling laws that only requires training a single model.

To begin, we assume that the goal of picking a fixed batch size $B$ is to minimize the L2 distance to the local CBS over the course of training. It is not obvious that minimizing L2 distance is the right way to pick the fixed CBS: for instance, we might want to weight training at a batch size *above* the local CBS more negatively than training below it. Regardless, we will proceed for now under the assumption that this is the right perspective. We also make the weaker assumption that $f(t) = 0$, in line with our empirical findings (Section 3). It follows that the best batch size to train at (i.e., fixed CBS) is simply the average local CBS over training:

**Proposition 1.** *Let $f(t)$ be integrable with $f(0) = 0$ and define*

$$R_2 = \sqrt{\int_0^T (B - f(t))^2 \, \mathrm{d}t}.$$

*Then $R_2$ is minimized by $B^* = \frac{1}{T} \int_0^T f(t)\mathrm{d}t$.*

*Proof.* We can first simplify the expression for $(R_2)^2$:

$$\begin{aligned}
(R_2)^2 &= \int_0^T (B - f(t))^2 \, \mathrm{d}t \\
&= \int_0^T \left(B^2 - 2Bf(t) + f(t)^2\right) \mathrm{d}t \\
&= B^2 T - \int_0^T \left(2Bf(t) - f(t)^2\right) \mathrm{d}t.
\end{aligned}$$

Now, taking the derivative with respect to $B$, we get

$$\frac{\mathrm{d}}{\mathrm{d}B}(R_2)^2 = 2BT - 2\int_0^T f(t)\mathrm{d}t.$$

Note that the second derivative $2T$ is positive. Thus, setting the derivative to $0$ and solving for $B$, we conclude that the following value of $B$ minimizes $R_2$:

$$B = \frac{1}{T}\int_0^T f(t)\mathrm{d}t. \qquad \square$$

Thus, under the assumptions we have made, if we are trying to pick a fixed batch size that best approximates the local CBS throughout training, we can simply pick the average CBS over training. We can use Proposition 1 to derive a scaling law for the fixed $B^*$ as a function of the final CBS or, equivalently, the total steps $T$. We now consider various reasonable functional forms $f(t)$ for the CBS.

### D.1 Power Law CBS Scaling

We first consider the prediction for the fixed CBS scaling law if the local CBS evolves as a power law.

**Proposition 2** ($B^*$ for power-law CBS). *Let $f(t) = t^c$ for $c > 0$. Then the fixed CBS is*

$$B^* = \frac{1}{c+1}T^c.$$

*Proof.* Plug in and solve the integral:

$$\begin{aligned}
B &= \frac{1}{T}\int_0^T t^c \mathrm{d}t \\
&= \frac{1}{T} \cdot \left[\frac{t^{c+1}}{c+1}\right]_0^T \\
&= \frac{T^c}{c+1}. \qquad \square
\end{aligned}$$

In the case where $c = 1/2$ (square root), $B^* = \frac{2}{3}B_T^* = \frac{2}{3}\sqrt{T}$, which derives the $\sqrt{T}$ scaling law proposed by prior work (Zhang et al., 2024).

| task | split | # shots | reference |
|------|-------|---------|-----------|
| ARC-Challenge | Test | 5 | (Clark et al., 2018) |
| ARC-Easy | Test | 5 | (Clark et al., 2018) |
| CommonsenseQA | Val | 5 | (Talmor et al., 2019) |
| HellaSwag | Val | 5 | (Zellers et al., 2019) |
| MMLU | Val and Test | 5 | (Hendrycks et al., 2021) |
| PIQA | Val | 5 | (Bisk et al., 2020) |
| Social IQa | Val | 5 | (Sap et al., 2019) |
| WinoGrande | Val | 5 | (Sakaguchi et al., 2020) |
| GSM8K | Gold | 5 | (Cobbe et al., 2021) |
| Minerva | Gold | 0 | (Lewkowycz et al., 2022) |
| Humaneval | Gold | 0 | (Chen et al., 2021) |
| MBPP | Gold | 0 | (Austin et al., 2021) |
| Copycolors 10-way | | 0 | (Wiegreffe et al., 2024) |

## D.2 Logarithmic CBS Scaling

**Proposition 3** ($B^*$ for log CBS). *Let $f(t) = \log(t+1)$. Then the fixed CBS is*

$$B^* = \frac{T}{T+1} \log(T+1) - 1.$$

*Proof.* Plug in and solve the integral:

$$
\begin{aligned}
B &= \frac{1}{T} \int_0^T \log(t+1)\mathrm{d}t \\
&= \frac{1}{T} \cdot \left[ ((t+1)\log(t+1) - t) \right]_0^T \\
&= \frac{T}{T+1} \log(T+1) - 1. \qquad \square
\end{aligned}
$$

Thus, for large $T$, the fixed CBS will scale as $B^* \approx \log T$.

## D.3 Discussion

These results show that, if we are choosing the fixed batch size to minimize average distance to the CBS as it evolves over training, we should pick it, more or less, as a simple function that slightly discounts the final CBS. Specifically, if we believe that the local CBS grows as $\sqrt{T}$ during training, then this derives the $\sqrt{T}$ scaling law for $B^*$ proposed in prior work.

One limitation of this view is that the L2 residuals may not be the right way to measure closeness to the CBS. In particular, it may be worse to overestimate the CBS compared to underestimate, as training above the CBS (with a scaled up learning rate) can be unstable. We thus do not read to much into this analysis, but view it as a potentially useful starting point for future empirical and theoretical that derives CBS scaling laws from the development of the local CBS over training.

## E   BPB Evaluation on Downstream Tasks

This section lists the datasets we used to compute BPB measures for downstream tasks. For multiple-choice tasks, we use the Cloze/Completion Formulation (CF), and compute the BPB metric on the gold answer. For completion tasks, we simply compute BPB over the correct answer. This approach was inspired by Bhagia et al. (2024). The selection of tasks follows the guidelines from Magnusson et al. (2025).

