# OpenReview forum: "Critical Batch Size Revisited: A Simple Empirical Approach to Large-Batch Language Model Training"
_NeurIPS.cc/2025/Conference — NeurIPS 2025 spotlight_

### Official Review · Reviewer_xcpd · 2025-06-28

**Clarity:** 2
**Significance:** 3
**Originality:** 3
**Rating:** 4
**Confidence:** 4

**Summary:**

This paper aims to answer the question that what’s the suitable batch size during training language models at scale. One existing work utilizes the gradient noise scale during training to estimate the critical batch size (CBS), below which training will not substantially degrade loss. However, this method requires two strong assumptions: SGD Optimizer and Well-Conditioned Optimization. In practice, these requirements are not typically met or be too expensive to realize, which limiting its applicability. To this end, this work introduces a simple, empirical approach to directly measure the CBS through branched training. The authors find that the CBS evolves over training, that is, CBS is near 0 at initialization, increases rapidly at first, and then plateaus as training progresses. Then, they present a natural way to reliably train language models at large batch size, i.e., batch size warmup method. Experiments with 1B- and 7B-size models demonstrate the effectiveness of the proposed approach.

**Questions:**

Line 58: but than -> but then?

**Ethical Concerns:**

["NO or VERY MINOR ethics concerns only"]

**Final Justification:**

I have a positive attitude toward the acceptance of this paper.

**Limitations:**

yes

**Quality:**

3

**Strengths And Weaknesses:**

Strengths:
1. They introduce an empirical method to directly measure the CBS via branched training.
2. They discover the finding that the CBS changes over the course of training, i.e., it improves rapidly initially but then flattens off. And, the CBS not depend strongly on model size.
3. They propose a batch size warmup strategy for large batch training: begin training with a small batch size and double it whenever the CBS increases sufficiency.

Weaknesses:
1. In table-1, compared to Large-Batch Control method, the proposed approach: Batch Size Warmup, gains less margin. For example, it only surpasses 0.7% in terms of PT Loss. It seems to be a performance fluctuate. Besides, it cannot save more gradient steps than Large-Batch Control method. In another word, it is better to use a larger batch size, e.g., 4096, to train a LM, rather than the proposed one.
2. The work claims that the proposed method utilizes 43% fewer gradient steps than the original training run. But in Lines 71-73, they also stated that their method is more expensive than launching many full training runs. These claims seem conflicting.
3. Does the data scale impact the CBS? In the experiments, 2B data is utilized. I wonder whether the CBS changes with another data size. Further, do different model architectures also impact the CBS? It lacks some discussion about that.
4. In the experiments, it lacks the comparison with the gradient noise scale method.
5. In the proposed batch size warmup strategy (Line 207), why double the current batch size? How to specify the hyper-parameter: 2? If we train another LM model with different model architecture (e.g., MoE),data scale, model scale, and optimizer, how to set it?

---

> ### Author Rebuttal · Authors · 2025-07-31
>
> Thank you for your review!
>
> Regarding weakness (1), our claim is that batch size warmup attains comparable (in some cases slightly better) loss compared to the small-batch run while using fewer gradient steps. On the other hand, the large-batch run degrades loss, as shown in Figure 4 and Table 2. The takeaway we want to highlight is that the warmup method is a way to achieve comparable loss with fewer gradient steps; while the large-batch method saves more gradient steps, we find it degrades loss.
>
> Regarding Lines 71-73, this was a typo about the branched training method. Thanks for catching this! In fact, branched training is *less expensive* compared to training many full runs, and batch size warmup does save 43% gradient steps relative to the small-batch control. We will update lines 71-73 to correctly say “less expensive”.
>
> Regarding data scale, we agree this is an interesting question, but outside the scope of this project, where we focus on the evolution of CBS over the course of training given a fixed data distribution. As mentioned in Section 2, studies [1] have suggested that CBS scales with data scale.
>
> Regarding the comparison of gradient noise scale with the branched training approach, the contrast between Figures 2 and 3 does give a comparison of these approaches for estimating the CBS. We do agree that it could have been interesting to consider batch size warmup with a gradient noise schedule, but we did not have time to launch this experiment during the rebuttal period, and, while it would have been interesting to have, this experiment is not central to the main claims in the paper (see further discussion in our response to Reviewer Hdu9).
>
> Regarding doubling the batch size, we chose this approach because of the practical constraint that we want the batch size to remain a multiple of the number of GPUs whenever we increase it. Because our number of GPUs during pretraining is a power of 2, doubling whenever we reach a new power of two was an easy way to maintain this invariant. In general, one could imagine more complex approaches, such as explicitly selecting the largest multiple of the number of GPUs less than the CBS. We will clarify our motivation and the potential alternatives around this hyperparameter choice.
>
> [1] https://arxiv.org/abs/2410.21676

---

> > ### Comment · Reviewer_xcpd · 2025-08-03
> > **Thanks for the response**
> >
> > The response has solved my concerns, and I have increased the score.

---

### Official Review · Reviewer_Hdu9 · 2025-06-30

**Clarity:** 4
**Significance:** 3
**Originality:** 3
**Rating:** 5
**Confidence:** 4

**Summary:**

This paper proposes and analyzes a method for estimating "critical batch size" (CBS). The paper measures the CBS by performing "branched training" for relatively small token windows, across multiple batch sizes, defining the CBS as the largest batch size at which the branched training does not reduce loss (with some tolerance/smoothing), as in Figure 1. The paper specifically compares this method to the gradient noise scale method of estimating CBS proposed by [McCandlish et al., 2018]. They point out that the gradient noise scale has specific, stringent underlying assumptions, and show empirically that it can underestimate CBS.

The paper then uses their branched training estimate of CBS to define a batch size warmup schedule. The authors show that when training a 1B parameter model, this slightly outperforms a small-batch size baseline in terms of loss and downstream evaluation metrics, while using significantly fewer gradient estimates. On the other hand, it does significantly better than a large-batch size baseline in terms of loss, while requiring somewhat more gradient steps.

**Questions:**

1. What happens when you use the CBS estimate from [McCandlish et al., 2018] to do batch size warmup, keeping everything else the same? That is, how much does the underestimate of the CBS from the gradient noise estimate translate into reduced performance on loss and/or downstream evaluation metrics?

2. How does the branched training estimate vary with the number of tokens $\Delta$ used for branched training? For example, would training with 1B tokens in Figure 1 give similar CBS estimates? What about 4B? You mention in L131-133 that larger $\Delta$ may produce larger CBS estimates - I don't immediately see the reason. Viewing CBS as a random function of $\Delta$, should we expect that its expected value changes with $\Delta$?

3. More generally, do you have any heuristic for selecting $\Delta$, potentially as a function of some model-scale-dependent training horizon (e.g. as some fraction of the Chinchilla-optimal number of training tokens)?

4. I found Appendix D to be quite interesting - is the reason it is not in the main body just due to lack of space? In the context of fitting such a power law, what would happen if you tried to fit your local CBS estimates to a power law? Do they obey this behavior, or do they exhibit qualitatively different behavior?

**Ethical Concerns:**

["NO or VERY MINOR ethics concerns only"]

**Final Justification:**

I have read all other reviewer responses and author feedback. Nothing about the strengths of the paper have changed in my opinion, and the other reviewers basically seem to have a positive impression as well.

**Limitations:**

Yes

**Quality:**

4

**Strengths And Weaknesses:**

## Strengths

I will be forthright: I think this is a well-written, well-executed, and novel paper, and should be accepted as such. Its clarity is perhaps its greatest virtue. It does a terrific job of being absolutely explicit, while still making the paper easy (even enjoyable) to read. I will also note that the paper bucks the trend of overstating contributions. The paper does a great job in its abstract and introduction in detailing exactly what they do. Their experimental setup is extremely clear, with explicit discussion of every experimental choice. I also think the paper does a good job of avoiding confusing, extraneous, or only tangentially related experimentation. Every figure and table has a clear purpose in contributing to the paper's overall message, and I would encourage the authors to continue to write papers like this.

The paper is sound, novel, and clearly relevant to a large body of work on pre-training dynamics. I will note that the paper's approach to measuring CBS is straightforward. I wish to fend off potential criticism on this front preemptively. This is a strength of the paper, not a weakness. The fact of the matter is that CBS has been an opaque phenomenon for years, and after reading this work I felt I understood it better. Not fully of course (the paper makes no mechanistic explanations for it, nor does it need to). But we can't study what we can't measure, and I'm particularly appreciative that the question of how to measure is approached so carefully. The approach leads to plenty of interesting open questions (some of which I detail below). Moreover, the empirics are well-done (with one caveat, as I discuss below) and make good use of datasets, models, and techniques training methodology that are widely used, increasing the paper's relevance.

## Weaknesses

Generally speaking, most my questions about the paper are in the vein of wanting to know more about the behavior of the branched training estimate. This is honestly a good critique to have - I have some questions that I think the authors could address, not because it's necessary for the paper to be accepted, but because I think the paper is quite compelling. That being said, there are a few issues that I want to emphasize.

### Ground-truth CBS

Figure 3 and the discussion around it (L173-180) is useful, but has an implicit assumption that I'm not sure I understand. The authors write that the gradient noise estimate of the CBS "underestimates the CBS by several orders of magnitude". What is the ground truth CBS that the authors are using in such a statement? Figure 3 would suggest that the answer is 4096 (at least, towards the end of training) but I'm not sure why this is treated as the actual CBS. This number is closer to what their own methodology suggests is the correct CBS, but that doesn't make it the ground truth. For example, is it possible that the method in Section 3.1 *overestimates* the CBS?

This lack of ground truth could be dealt with another way, via downstream evaluation, which unfortunately brings me to my next point.

### Why not train with a gradient noise scale-based batch size schedule?

I think that Figure 3 is somewhat, but not wholly convincing in terms of the superiority of the method in 3.1 to the gradient noise scale method of estimating CBS. To that end, I'm surprised that the authors didn't train a model using a batch size schedule derived from the gradient noise scale CBS. The warmup could be the same as in 4.1, just change the CBS estimator. If (and I suspect it would) the gradient noise scale yields an inferior model to the batch size based on branched training, that'd be a great point of evidence in favor of branched training! Regardless of being accepted, this is something that I hope the paper includes at some point - I think it really would make the comparison point much clearer.

### How does the CBS estimate depend on $\Delta$?

This is a more minor issue, but one that was on my mind while reading. The branched training depends on the window size $\Delta$. L128-L133 discusses the role of $\Delta$ more briefly than I wanted. For example, the statement that larger $\Delta$ may yield larger CBS estimates makes some sense, but it'd be nice to see some kind of exploration of this fact. For example, would a miscalibrated $\Delta$ yield better or worse CBS estimates than gradient noise scale estimates? Moreover, how would the CBS estimate change between settings where $\Delta$ is too small versus settings where $\Delta$ is too large?

I will note that no paper can provide comprehensive answers to everything. I think it's reasonable to treat this specific "critique" as evidence for accepting the work - the questions are natural ones that could (and should) be answered by follow-up work.

## Other comments

I have left other comments on the work, some overlapping with the above, some not, in the form of questions below.

---

> ### Author Rebuttal · Authors · 2025-07-31
>
> Thank you for your review and your appreciation of our work! We are very happy to hear that you appreciated the contributions and clarity of the paper.
>
> Regarding your questions about the branched training method, we have some thoughts.
> > The authors write that the gradient noise estimate of the CBS "underestimates the CBS by several orders of magnitude". What is the ground truth CBS that the authors are using in such a statement? Figure 3 would suggest that the answer is 4096 (at least, towards the end of training) but I'm not sure why this is treated as the actual CBS. This number is closer to what their own methodology suggests is the correct CBS, but that doesn't make it the ground truth. For example, is it possible that the method in Section 3.1 overestimates the CBS?
>
> This is a good question, and we will try to make our reasoning more explicit. Indeed, the comparison between Figures 2 and 3 suggests that the gradient noise scale underestimates the CBS relative to our method. A priori, it is possible that our method is an overestimate and that the gradient noise scale is correct. However, our pretraining results in Figure 4 provide evidence against this (i.e., they show that our method is not an overestimate), because we can train with the larger batch size predicted by our method and obtain no degradation in final loss. By definition of the CBS, this means the batch size produced by our method does not exceed the CBS (and thus the smaller gradient noise estimate is an underestimate).
>
> > Why not train with a gradient noise scale-based batch size schedule?
>
> We think this could have been a nice, though not entirely central, comparison to have. The prediction based on the previous sections would be that we could still obtain decent pretraining loss (since we’re training below the CBS) but that the number of gradient steps would be significantly larger than training with the branched training batch size. The latter claim is true by construction and thus not worth testing, but it could have been interesting to measure the loss attained. On the other hand, it takes a while to launch the pretraining runs behind the results in Figure 3, and we were not able to launch this pretraining run during the short rebuttal period, especially given that the results would not be essential to our main findings.
>
> > How does the CBS estimate depend on $\Delta$?
>
> Under the local recovery assumption (Assumption 3), increasing $\Delta$ up to the total pretraining token budget will only increase the estimate of the CBS, so CBS estimates with smaller $\Delta$ are still a lower bound. The choice of $\Delta$ can thus be understood as striking a tradeoff between finding the largest possible CBS lower bound and the compute budget one is willing to spend (since larger values of $\Delta$ require more training). We did not explore this tradeoff in further empirical detail since we chose to allocate our attention and compute towards the large scale batch size warmup pretraining runs instead. However, we’ll try to clarify the role of $Delta$ in the paper and add a note that understanding the role of $\Delta$ is an open question for future work.
>
> > I found Appendix D to be quite interesting … In the context of fitting such a power law, what would happen if you tried to fit your local CBS estimates to a power law?
>
> We spent some time trying to fit a parametric form to our CBS plots, and we did not find any form that we were confident fit the shape of the function. This could be because there is a phase change between the first and second parts of the trend. Because we did not have any conclusive insights here, we did not make any claims about the functional form of the CBS trend in the paper, though we thought it would still be interesting to include the theoretical investigation related to this in the appendix since it might be relevant for future research.

---

> > ### Comment · Reviewer_Hdu9 · 2025-07-31
> > **Thanks for the response**
> >
> > This all makes sense to me. I wish to say that I 100% agree with not running more experiments during the rebuttal phase.
> >
> > I do want to mention that your comments about (I) ground-truth CBS and (II) non-centrality of running experiments against the gradient noise scale batch size schedule are somewhat in tension. If the ground-truth is validated by the CBS, then it'd be useful (even if unsurprising!) to have a matching result on the gradient noise scale.
> >
> > I mention this on the off chance that this is possible to do before the camera-ready, but I don't consider this blocking by any means.
> >
> > Last, I want to clarify that my question about CBS and its dependency on $\Delta$ is about whether or not we should believe that CBS is "unbiased" with respect to $\Delta$ or not (e.g. if I plotted CBS with respect to $\Delta$, should I expect it to look approximately flat, within some decreasing noise threshold?). But this is somewhat of an academic question.

---

### Official Review · Reviewer_6mhC · 2025-07-02

**Clarity:** 4
**Significance:** 3
**Originality:** 3
**Rating:** 5
**Confidence:** 4

**Summary:**

This paper studies "critical batch size" (CBS), the largest batch size a training run can use without hurting the performance (assuming hyperparameters like learning rates are adjusted correctly). Prior methods use gradient noise to estimate CBS, but it is based on the assumption of using SGD and well-conditioned optimization, which is unrealistic.

The three main contributions of this paper is very clear:

(1) This paper proposes a new empirical way to estimate CBS by (a) starting from a checkpoint in the middle of training and (b) training for a short period of time and checking if the performance matches the baseline. Compared to prior method, this only needs to train the model for a little bit. It also allows for discovering the local CBS, while prior method can only find a global CBS.

(2) By using the aforementioned method, the authors discovered that CBS is small at the beginning of training, then increases, and then plateaus. The final CBS does not differ much across scales (the authors used 1b and 7b).

(3) To accommodate the finding mentioned before, the authors proposed a batch size warmup regime (double the batch size every time CBS increases). The authors showed that this leads to slightly better performance than using either the initial (smallest) or the critical (largest) batch size throughout the training.

**Questions:**

From Figure 2, it seems that the 1B model should reach the batch size of 4K at 100B tokens? However, the authors say that this is reached at 503B tokens in section 4. Not sure where I get it wrong?

**Ethical Concerns:**

["NO or VERY MINOR ethics concerns only"]

**Final Justification:**

I'm very happy with the author response and will maintain my positive score.

**Limitations:**

Yes

**Quality:**

3

**Strengths And Weaknesses:**

**Strength**

The new way of empirically finding CBS is novel and allows us to examine the local CBS at any given point of training. The finding that CBS changes throughout training is super interesting and potentially reveals how gradient noise changes throughout training as well. The batch size warmup regime, though not significantly better than using one CBS throughout training, is something.

**Weakness**

My main complain is on the experiments with the batch size warmup regime. First, there should be more evaluation (for example, few-shot downstream tasks evaluation). The authors used the loss on these tasks, which is not as straightforward to interpret. Second, if I understand correctly, the schedule is based on the results from Section 3. It would be more complete of a method if the authors propose a way to online check the CBS inflection point.

---

> ### Author Rebuttal · Authors · 2025-07-31
>
> Thanks for your review!
>
> Regarding the downstream evaluations, we have found the loss-based evaluations as represented in https://arxiv.org/abs/2412.04403 to be some of the most reliable for evaluating pretraining interventions. Of course, this kind of evaluation is an unsolved problem with a lack of scientific consensus. If you have some specific suggestions for alternate types of evaluations we could use here, we would be curious for suggestions.
>
> Indeed, the batch size warmup schedule is based on the results in Section 3. Because we only needed to determine two thresholds for increasing the batch size, we determined them manually based on Figure 2 before launching the batch size warmup pretraining runs. We did not have the resources to relaunch the pretraining runs using a more online approach, but we agree this would be a useful refinement of the approach and revise the paper to clarify this.
>
> > From Figure 2, it seems that the 1B model should reach the batch size of 4K at 100B tokens? However, the authors say that this is reached at 503B tokens in section 4. Not sure where I get it wrong?
>
> We chose the thresholds manually and conservatively in Figure 4, with the goal of having the warmup batch size upper bounded by the curve in Figure 2. Based on a noisier initial version of Figure 2, we preferred a later threshold for increasing to batch size 4K, and once we produced the newer Figure 2 plot, we did not have the resources to relaunch the pretraining runs. We will clarify this manual choice in the paper and emphasize that there is an opportunity for future work to determine batch size thresholds more systematically.

---

> > ### Comment · Reviewer_6mhC · 2025-08-04
> > **Thanks for the response**
> >
> > Thanks for the response and I will maintain my positive score.

---

### Official Review · Reviewer_AKeu · 2025-07-06

**Clarity:** 4
**Significance:** 3
**Originality:** 3
**Rating:** 5
**Confidence:** 4

**Summary:**

This paper presents a practical method to empirically determine the **critical batch size** in large-scale language model pretraining—the largest batch size at which performance does **not** degrade compared to training with smaller batches and more steps. Beyond this threshold, further increasing batch size while reducing the number of steps begins to hurt final model quality.

The authors introduce a simple on-the-fly test for identifying this threshold: take a checkpoint from an ongoing training run, resume training with a larger batch size and appropriately scaled learning rate (e.g., following square-root scaling), and observe the smoothed training loss after a short additional training segment. If the loss remains within a fixed tolerance (e.g., within 0.01 of the loss without changing the batch size), the new batch size is deemed acceptable.

Using this method, they analyze the OLMo-1B and OLMo-7B models and find that the critical batch size increases sharply in the early stages of training (for around 200B) but plateaus later. The analysis also shows that **Gradient Noise Scale**—a previously proposed heuristic for estimating the largest usable batch size underestimates the critical batch size by roughly a factor of two, making it overly conservative.

Based on these findings, the authors propose a batch size warm-up strategy: begin training with a smaller batch size and increase it gradually during training, adjusting the learning rate accordingly. They decide the steps to increase the batch size by referring to the complete OlMo 1B run. This strategy improves training efficiency by reducing the number of steps needed to reach target loss while maintaining or slightly improving downstream performance compared to a small batch size baseline.

**Questions:**

1. Is the batch size set to a constant in pretraining and mid-training in these experiments? Is there a possibility that it should be different at the end of pretraining versus at the beginning of mid-training?

2. In terms of wall time, how much saving can the batch size warmup produce?

**Ethical Concerns:**

["NO or VERY MINOR ethics concerns only"]

**Final Justification:**

This paper offers a clean way to specify the critical batch size in pretraining and I don't see any significant issues about the submission.

**Limitations:**

yes

**Paper Formatting Concerns:**

I didn't notice any.

**Quality:**

4

**Strengths And Weaknesses:**

**Strengths:**

1. The proposed method is simple, easy to understand, and has minimal assumptions, making it a convincing method to determine CBS in practice.

2. The paper provides strong and thorough empirical evidence to support their findings on the critical batch size, showing that their method effectively improves training efficiency.

3. Overall, the paper is clearly written and easy to follow.

**Weaknesses:**

1. The authors didn't discuss a related work on optimal learning rate and batch size scaling for Adam [1], which generalizes the gradient noise scale method. Including a discussion or comparison with this related work would have strengthened their argument.

2. There seems to be some minor arbitrariness in how the thresholds for increasing the batch size are set. In their experiments, the critical batch size appeared to be around 4096 as early as 200 billion tokens, but the authors only set the threshold to increase the batch size at 600 billion tokens, which is much later in the training process.

[1]  Surge Phenomenon in Optimal Learning Rate and Batch Size Scaling

---

> ### Author Rebuttal · Authors · 2025-07-31
>
> Thanks for your review! We are glad to hear that you find our paper well-written and that its findings are convincing.
>
> We’ll make sure to incorporate some comparison with [1] as you suggest.
>
> Regarding the minor arbitrariness in setting the batch sizes thresholds for the warmup experiment, we set the threshold manually early on the research project based on a preliminary version of Figure 2 and then launched the batch size warmup pretraining runs. We agree that, all else equal, it would be nice to eliminate this arbitrariness. However, the pretraining runs for the warmup experiments were quite expensive, and we judged that the substantial cost for rerunning those experiments was not worth it to eliminate minor arbitrariness, especially since we believe the results are not especially sensitive to a precise choice of threshold. We will add some brief discussion of this limitation in the paper and highlight that making the choice of threshold more systematic would be appealing in future work.

---

> > ### Comment · Reviewer_AKeu · 2025-08-01
> >
> > I have the rebuttal and will keep the positive rating.

---

### Decision · Program_Chairs · 2025-09-17

**Decision:**

Accept (spotlight)

**Comment:**

This paper proposes a new method for estimating the critical batch size in LLMs that avoids restrictive assumptions and shows how it will changes over the course of training. The generality of the method to MoE’s or beyond LLMs is not studied and there were some concerns on the warmup scheduled. Overall the work is clearly written, easy to follow, and backed by experiments on realistic model scales. The reviewers were unanimous about accept.